# Randomized Controlled Trials to Treat Obesity in Military Populations: A Systematic Review and Meta-Analysis

**DOI:** 10.3390/nu15224778

**Published:** 2023-11-14

**Authors:** Davide Gravina, Johanna Louise Keeler, Melahat Nur Akkese, Sevgi Bektas, Paula Fina, Charles Tweed, Gerd-Dieter Willmund, Janet Treasure, Hubertus Himmerich

**Affiliations:** 1Centre for Research in Eating and Weight Disorders, Institute of Psychiatry, Psychology and Neuroscience, King’s College, London SE5 8AF, UK; johanna.keeler@kcl.ac.uk (J.L.K.); melahat_nur.akkese@kcl.ac.uk (M.N.A.); sevgi.bektas1@kcl.ac.uk (S.B.); janet.treasure@kcl.ac.uk (J.T.); hubertus.himmerich@kcl.ac.uk (H.H.); 2Department of Clinical and Experimental Medicine, University of Pisa, 56127 Pisa, Italy; 3South London and Maudsley NHS Foundation Trust, Bethlem Royal Hospital, Monks Orchard Road, Beckenham BR3 3BX, UK; charles.tweed@slam.nhs.uk; 4Department of Psychology, Hacettepe University, Ankara 06800, Türkiye; 5Faculty of Psychology, Sigmund Freud University Vienna, Freudplatz 1, 1020 Vienna, Austria; paula@thefinas.eu; 6Royal Navy Reserve, London WC1N 1NP, UK; 7Bundeswehr Center for Military Mental Health, Military Hospital Berlin, 13, 10115 Berlin, Germany; gw@ptzbw.org

**Keywords:** obesity, obesity treatment, weight loss intervention, military population, active-duty military personnel, veterans, RCT, randomized controlled trial, meta-analysis

## Abstract

In recent years, overweight and obesity have reached an alarmingly high incidence and prevalence worldwide; they have also been steadily increasing in military populations. Military personnel, as an occupational group, are often exposed to stressful and harmful environments that represent a risk factor for disordered eating, with major repercussions on both physical and mental health. This study aims to explore the effectiveness of weight loss interventions and assess the significance of current obesity treatments for these populations. Three online databases (PubMed, PsycInfo, and Web of Science) were screened to identify randomized controlled trials (RCTs) aiming to treat obesity in active-duty military personnel and veterans. Random-effects meta-analyses were conducted for body weight (BW) and body mass index (BMI) values, both longitudinally comparing treatment groups from pre-to-post intervention and cross-sectionally comparing the treatment group to controls at the end of the intervention. A total of 21 studies were included: 16 cross-sectional (BW: n = 15; BMI: n = 12) and 16 longitudinal (BW: n = 15; BMI: n = 12) studies were meta-analyzed, and 5 studies were narratively synthesized. A significant small overall BW and BMI reduction from baseline to post-intervention was observed (BW: *g* = −0.10; *p* = 0.015; BMI: *g* = −0.32; *p* < 0.001), together with a decreased BMI (*g* = −0.16; *p* = 0.001) and nominally lower BW (*g* = −0.08; *p* = 0.178) in the intervention group compared to controls at the post-intervention time-point. Despite limitations, such as the heterogeneity across the included interventions and the follow-up duration, our findings highlight how current weight loss interventions are effective in terms of BW and BMI reductions in military populations and how a comprehensive approach with multiple therapeutic goals should be taken during the intervention.

## 1. Introduction

Obesity is a global epidemic. Overweight and obesity are currently two of the main public health concerns across the world, with more than 1 billion people worldwide being obese—650 million adults, 340 million adolescents, and 39 million children [1]. Worldwide obesity has nearly tripled since 1975, and this number is still dramatically increasing. According to the World Obesity Atlas report from the World Obesity Federation, the majority of the global population (51%, or over 4 billion people) will be living with either overweight or obesity by 2035 based on current trends, and 1 in 4 people worldwide (nearly 2 billion) will meet or exceed a body mass index (BMI) of 30 kg/m^2^ [2].

Moreover, obesity is a systemic disease impacting the well-being of the person as a whole, with consequences for both physical and mental health. Indeed, the World Health Organization (WHO) estimates that by 2025, approximately 167 million people—adults and children—will become less healthy because they are overweight or obese [1]. Overweight and obesity are major risk factors for many chronic diseases, including cardiovascular diseases such as hypertension, dyslipidemia, heart diseases, and stroke, which are the world’s leading causes of death [3]. Being overweight can also lead to type 2 diabetes and musculoskeletal disorders such as osteoarthritis and represents a risk factor for the occurrence of some tumors, including endometrial, breast, ovarian, prostate, liver, gallbladder, kidney, and colon [1].

Obesity is one side of the double burden of nutritional problems, affecting all age groups and countries world-wide regardless of their developmental stage [4,5,6,7]. Today, more people are obese than underweight in every region of the world except sub-Saharan Africa and Asia [1]. As mentioned above, the issue has grown to epidemic proportions, with over 4 million people dying each year because of being overweight or obese and its consequences [1].

It is now widely accepted that occupational factors may play an important role in the occurrence of excessive body weight [8,9,10]. The main occupational risk factors are performing sedentary work and adverse lifestyle factors such as high levels of occupational stress [8,11,12,13]. Military personnel represent a population particularly exposed to a higher level of stress, with a higher risk of both obesity and mental health conditions such as major depression, post-traumatic stress disorder (PTSD), and disinhibited eating [14]. 

Indeed, the obesity epidemic has reached the military population [15,16]: possibly related to high levels of stress and harmful environmental factors, especially during military exercises, military missions, or during deployment and relocation [8,17,18]. In addition, there may be limitations on food selection or availability, especially among particular services such as the Navy and Marine Corps [19].

Another consideration is that military personnel are required to meet body weight and composition standards to remain in the military and to be eligible for their work. This might lead to an increased focus and concern on body weight, shape, and fitness. In fact, soldiers are required to maintain fitness standards that are not required for civilians to be ready for combat. Nonetheless, the armed forces population is experiencing similar patterns of increasing levels of overweight and obesity as observed in civilian society [20,21].

Between 2002 and 2015, the rate of obesity among U.S. active-duty military service members increased by 68%, such that now nearly two-thirds of U.S. active-duty across all branches meet criteria for overweight and obesity [15,22]; and, thereafter, the overall prevalence of obesity within the U.S. active component increased from 16.3% in 2015 to 17.9% in 2019 [23,24]. Obesity was significantly higher among Navy and Air Force military personnel [15], In particular, Navy personnel have the third highest rate of overweight and obesity among all service branches (64.6%), with 48.9% overweight and 15.7% obese [22]. Evidence from the UK shows slightly lower collective values, with 38% being overweight and 14% obese [20]. In the scientific literature, soldiers fulfilling the criteria of obesity constitute more than 15% in the US Army [23,25], 12% in the British Army [26,27], 13% in the Iranian Army [28,29], 6% in the Polish Air Force [30], and 44% in the Saudi Arabian Army [31].

Moreover, the spread of the COVID-19 pandemic in March 2020 led to restrictions, which may have contributed to a worsening problem of obesity within the active military. A recent study by Legg et al. [23] reported that the monthly prevalence of obesity in U.S. active component military members ranged from 15.0% in August 2020 to 19.3% in April 2021, confirming a further growth trend. Again, among the services, the Navy and Marine Corps showed the largest absolute increase in mean monthly obesity prevalence from the pre-pandemic period to the pandemic period (0.78% and 0.77%, respectively) [23].

Not only does obesity within the military ranks negatively impact the professional perception of the military in terms of appearance, but it also compromises function. The psychological and physiological impacts on obese military personnel include problems with cardiorespiratory fitness and neuromuscular fitness [32], heat stress [33], sleep apnea [34], a higher risk of musculoskeletal injuries and load carriage [35,36], and also mental health problems such as depressive symptoms [37], PTSD [38], anxiety, and substance and alcohol abuse disorders [39,40].

Veterans also have higher levels of obesity: 78% of U.S. military veterans are overweight or obese [41], and 65% of female and 45% of male veterans report at least one symptom of Binge Eating Disorder (BED) [42]. Sedentary habits and suboptimal levels of physical activity [43] may combine with PTSD (seen in 11 and 30% of veterans) to increase disordered eating behaviors [44,45]. Thus, addressing eating behavior within this population is a high priority [46].

A variety of weight management programs have been developed to address the problem of obesity in military personnel, including prevention programs [47], lifestyle programs such as “LOOK AHEAD” [48,49], “LE3AN” [50], “MOVE”, and “armyMOVE!” [51,52], nutrition-focusing programs [53,54], internet-based interventions [55,56,57], cognitive behavioral interventions [58,59], Navy weight management programs [60], pharmacological interventions [61], and surgical approaches [62,63]. 

Several systematic reviews of weight management programs have been undertaken. A systematic review in 2011 by Sanderson et al. [20], including 17 studies, found that interventions based on exercise, healthy eating information, behavioral modification, and structured follow-up were effective for weight reduction. Moreover, a systematic review conducted in 2017 summarized data from 38 studies and found weight loss for up to 12 months from dietary, physical activity, and weight management interventions among active-duty military personnel [64]. However, a systematic review in 2021 of seven studies that evaluated the effectiveness of weight loss interventions among U.S. active-duty military populations concluded that there is not a sufficient body of evidence to determine if interventions are effective [65].

At the time of writing this paper, there has been no meta-analysis published regarding weight management programs in military populations. Therefore, the present systematic review and meta-analysis aims to synthesize the results of studies investigating obesity treatments in military populations, in which the interventions have been delivered with a randomized controlled trial study design.

The main aim of the study is to address the effectiveness of the therapeutic intervention in terms of weight loss by comparing the pre-intervention body weight and BMI values of participants who received the treatment to the post-intervention values, as well as comparing the intervention group to controls.

A secondary aim of this study is to investigate the extent to which demographic and intervention characteristics, including age, duration of the intervention, body weight (BW) at baseline, and body mass index (BMI) at baseline, may be associated with changes in weight.

## 2. Materials and Methods

This systematic review and meta-analysis was registered in PROSPERO (CRD42023439107) and was conducted according to the Preferred Reporting Items for Systematic Reviews and Meta-Analyses (PRISMA) guidelines [66].

### 2.1. Search Strategy

Two reviewers (DG and PF) independently and systematically searched the following electronic databases: Web of Science, PubMed, and APA PsycInfo (Ovid) from inception until 7 September 2023. Searches included the following keywords: “obesity” or “adiposity” or “overweight” in combination with “military” or “military personnel” or “army” or “navy” or “military force” or “air force” or “soldiers” and “treatment” or “clinical trial” or “psychotherapy” or “psychological therapy” or “CBT” or “cognitive behavioral therapy” or “pharmacological” or “medication” or “drug”. The full search strategy is listed in Appendix A. Searches were supplemented by internet searches and manual hand searches through reference lists to identify potentially relevant additional studies.

### 2.2. Eligibility Criteria

To be included in the systematic review, studies needed to meet the following criteria:➢Human studies➢The studies included only individuals aged ≥ 18 years➢The studies involved military populations: Army, Navy, or Air Force personnel, Active duty-military personnel, veterans➢The topic of the studies focused on weight management interventions (any kind of treatment, e.g., pharmacological, psychotherapeutic, lifestyle, and nutritional) to treat obesity and overweight➢The studies assessed randomized controlled trials (RCTs) to test the treatment➢The studies are original articles➢The studies are published in the English, German, Italian, Spanish or French language ➢Ongoing studies were eligible in order to maximize inclusion➢Articles from literature that met the following criteria were eligible for exclusion:➢The topic is not related to weight management interventions or obesity treatment➢The sample is different from the military population➢The study design is different from a randomized controlled trial➢Articles not published in English, German, Italian, Spanish, or French. ➢Animal or pre-clinical studies➢Article type other than original articles (i.e., systematic reviews, narrative reviews, meta-analyses, cross-sectional studies, perspective papers, letters (without data), masters or doctoral theses, case reports).

### 2.3. Study Selection

The search process was conducted independently by two reviewers (DG and PF). Titles and abstracts of publications yielded by the searches were imported into EndNote, where duplicates were removed. Titles and abstracts of the remaining records were imported into Rayyan and assessed against the aforementioned eligibility criteria. Those deemed highly unlikely to be relevant were discarded. Full texts of the remaining articles were assessed against the eligibility criteria before inclusion (see Figure 1). A third reviewer (HH) supervised the process and resolved any disagreements.

### 2.4. Data Extraction

The first author (DG) extracted data from all included studies into an electronic summary table, which was checked by another reviewer (HH). The following data were extracted: publication identifiers (journal, year, first author); country of origin; study design; study objective; methodology; sample characteristics (mean age, sample size, body weight, BMI); clinical characteristics (type of intervention, duration of the intervention, main findings, outcome data). Authors were contacted for data not obtainable from the manuscript.

### 2.5. Risk of Bias Assessment

Two reviewers inspected each study to assess the risk of bias independently by using the Scottish Intercollegiate Guidelines Network (SIGN) Methodology Checklist for randomized controlled trials (Appendix A), evaluating the internal validity of the RCTs and the overall assessment of the studies through several domains of bias. A third reviewer coordinated the process and resolved any disagreements. See Appendix A for the assessments of study quality.

### 2.6. Data Synthesis and Statistical Analysis

Individual meta-analyses were conducted in STATA (Release 17; StataCorp LP) using the ‘meta-set’ and ‘meta-summarize’ commands. The primary outcome measures were body weight (kg) and BMI (kg/m^2^). Higgins *I*^2^ was used to assess between-study heterogeneity, which was considered to be high when *I*^2^ > 75%. A random-effects model using the DerSimonian and Laird method [67] was used to pool the effect size (ES; Hedge’s *g*) from the between-group difference for each study (Interventions vs. Controls, and Pre vs. Post) relative to the sample size. The ES expressed the difference between BW and BMI at baseline and at the last available follow-up timepoint for longitudinal meta-analyses (pre-to-post intervention). Regarding the cross-sectional meta-analyses, the ES expressed the difference between BW and BMI at post-intervention, where values were compared between treatment and control groups. Statistical significance of between-group differences was ascertained according to the *p* < 0.05 threshold.

We performed meta-regressions, using the ‘metareg’ command, to investigate the association between key demographic, study, and clinical variables (BMI at baseline, BW at baseline, mean age, duration of the intervention), and the ES of the meta-analysis comparing the pre-to-post intervention groups. The meta-regression analyses were not limited by the number of studies, as it is recommended that meta-regressions be conducted when there are 10 or more studies available for inclusion [68]. 

Alongside the evaluation of the primary outcome of the study, additional meta-analyses divided by sample and intervention type were conducted using the ‘meta-set’ and ‘meta-summarize’ commands, with the aim of evaluating the effectiveness of the different types of interventions included in the study.

Publication bias was assessed using Egger’s test for small study effects, with funnel plots derived using the ‘meta funnelplot’ command. The Duval and Tweedie Trim and Fill method [69] was used successively if funnel plot asymmetry was detected to identify whether there were any missing studies.

## 3. Results

The literature search resulted in 3105 studies (Figure 1). After removing duplicates and reviewing titles and abstracts, 80 studies were retrieved for full-text review. A further 65 studies were excluded for the following reasons: ineligible publication format (n = 13), lack of a control group (n = 34), and not a military sample (n = 18). Finally, 21 studies [57,60,70,71,72,73,74,75,76,77,78,79,80,81,82,83,84,85,86,87,88] were included in this systematic review, 16 of which [57,71,72,74,75,76,77,78,80,81,82,83,84,85,87,88] were included in the meta-analysis.

### 3.1. Characteristics of Included Studies and Participants

A total of 4253 participants from 21 studies were included in this systematic review and meta-analysis. All studies had a randomized controlled clinical trial design. The duration of the interventions ranged from two weeks to twenty-four months, with a median of 26 weeks. The average age of participants ranged from 21.0 years to 61.8 years (mean ± SD: 45.6 ± 15.8), and most of the overall sample was represented by males (58%). The overall mean ± SD BW was 97.4 ± 21.2, and the overall mean ± SD BMI was 32.4 ± 5.6. The majority of the studies were conducted in the United States of America (n = 18); two studies were from Iran and one from Brazil. The military population studied included active-duty military soldiers (n = 7), Navy personnel (n = 3), Air Force personnel (n = 3), and veterans (n = 8).

Of these 21 included studies, sixteen studies [57,71,72,74,75,76,77,78,80,81,82,83,84,85,87,88] were included in the longitudinal meta-analyses investigating the effect sizes of BW [71,72,74,75,76,77,78,80,82,83,84,85,87,88] and BMI [57,71,72,77,78,80,81,82,83,84,85,87] comparing pre- to post-intervention values for the treatment group (11 studies were used in both the BW and BMI longitudinal meta-analyses). Sixteen studies [57,71,72,74,75,76,77,78,80,81,82,83,84,85,87,88] were also included in the cross-sectional meta-analyses comparing BW [57,71,72,74,75,76,77,78,80,82,83,84,85,87,88] and BMI [57,71,72,77,78,80,81,82,83,84,85,87] data from the treatment group to controls at the post-intervention time-point (11 studies were used in both the BW and BMI cross-sectional meta-analyses).

Regarding the longitudinal meta-analyses, the pooled mean ± SD age for the intervention group was 42.5 ± 15.8 for the meta-analysis investigating BW (reported by 12 studies) and 35.8 ± 11.7 for the meta-analysis investigating BMI (reported by 9 studies). The pooled mean ± SD BW was 93.8 ± 20.7 for the pre-intervention group and 91.9 ± 20.0 for the post-intervention group, as reported by 15 studies. The pooled mean ± SD BMI reported by 12 studies was 30.6 ± 5.2 and 29.9 ± 5.2 for the pre-intervention and post-intervention groups, respectively. 

Regarding the cross-sectional meta-analyses, the pooled mean ± SD age was 42.5 ± 15.8 for the treatment group and 44.0 ± 16.2 for the control group (reported by 12 studies) in the meta-analysis investigating BW; the pooled mean ± SD age for the cross-sectional meta-analyses investigating BMI was 35.8 ± 11.7 for the treatment group and 35.9 ± 11.6 for the control group (reported by 9 studies). The pooled mean ± SD BW was 91.9 ± 20.0 in the treatment group and 95.3 ± 19.9 in the control group, as reported in 15 studies. The pooled mean ± SD BMI was 29.9 ± 5.2 and 30.7 ± 5.2 for the treatment and control groups, respectively, reported in 12 studies.

Findings from five studies [60,70,73,79,86] were narratively synthesized and not included in the meta-analysis due to the required data being unavailable (n = 3) or because the studies were still in progress (n = 2). The characteristics of each study are summarized in Table 1.

### 3.2. Weight Loss Interventions of the Included Studies

Across 21 studies, the majority of papers (n = 19) reported an intervention based on cognitive behavioral therapy or behavioral modification, except for two studies [81,84]. 

Diet education or nutritional modifications was offered by twelve of the interventions [57,72,73,75,78,80,81,83,84,85,87,88], as well as one study that provided a synbiotic supplementation [81] and two studies that provided a meal-replacement program [78,84]. In twelve studies [57,60,70,71,72,76,77,82,83,86,87,88], participants self-reported outcomes such as weight, whereas in twelve studies [60,70,71,73,74,75,78,79,80,81,84,85], specialists were available for monitoring, counselling, and outcome measurement.

Specific devices, such as a pedometer [86] and a calorimeter [80], were used in two studies, and in five studies, the interventions were delivered through an Internet-based program or Internet-assistance were provided during the intervention [57,76,78,86,87].

Additionally, two studies used a specific weight loss program for the Navy population, or Marines [60,72]. One study focused on a post-partum weight loss program [83], and another study explored a weight loss maintenance program to address the amount of weight regain and retention during follow-up [88].

Finally, one study investigated the efficacy of the pharmacologic treatment Orlistat versus placebo [85].

A summary of the interventions provided by each study is shown in Appendix A.

### 3.3. Results for the Meta-Analysis Comparing Pre-to-Post Intervention for the Treatment Group

Data from a total of 15 studies, including a total sample of 2666 participants (pre-intervention group n = 1431; post-intervention group n = 1235) were used to compare the pre-treatment values of the BW of participants who received the intervention with the post-treatment values. Similarly, a total sample of 1942 participants from 12 studies was used to compare BMI values (pre-intervention group n = 1028; post-intervention group n = 914). Forest plots for the BW and BMI meta-analyses are presented in Figure 2 and Figure 3, respectively. A summary of comparative outcomes and heterogeneity for BW and BMI meta-analyses for the intervention group are shown in Table 2.

Participants who received the intervention showed overall lower post-treatment BW values than pre-treatment, with a small but significant effect (Hedges *g* = −0.10; 95% CI −0.02, −0.18; *p* = 0.015). Furthermore, results showed intervention effectiveness also in terms of BMI reduction, as a small-to-medium but significant overall reduction of post-treatment body mass index than baseline was observed (*g* = −0.32; 95% CI −0.15, −0.48; *p* = 0.0002).

#### Results for the Meta-Analyses Comparing Pre-to-Post Outcomes Per Sample Type

Additional meta-analyses were conducted, separating active military personnel from veterans, with the aim of evaluating the effectiveness of the weight loss interventions on these different samples analyzed individually (Table 2). Results showed a significant decrease in both BMI (*g* = −0.35; *p <* 0.001) and BW (*g* = −0.12; *p* = 0.041) in active-duty military personnel. In veterans, there was a small decrease in BW, although this was not significant (*g* = −0.09; *p* = 0.173). There was insufficient data to meta-analyze outcomes relating to BMI in veterans. 

### 3.4. Results for the Treatment Group vs. Controls Meta-Analyses

A total of 15 studies were included in the meta-analysis comparing BW data for the interventional group to the comparison group, with a total sample of 2329 participants (intervention group, n = 1235; control group, n = 1094). Results showed an overall lower body weight in the treatment group compared to controls at the post-intervention time-point (T1), although statistical significance was not achieved (*g =* −0.08; 95% CI −0.19, 0.03; *p* = 0.178). The forest plots for BW and BMI are presented in Figure 4 and Figure 5, respectively, and comparative outcomes and heterogeneity are summarized in Table 2.

A total sample of 1666 participants from 12 studies was used to compare BMI values (intervention group n = 914; control group n = 752). At post-intervention, a significantly lower BMI in the treatment group was found in comparison with controls (*g =* −0.16; 95% CI −0.26, −0.06; *p* = 0.001). 

#### Results for Treatment Group vs. Control Meta-Analyses Per Sample Type

Results showed that active-duty personnel in the treatment groups had a significantly lower BMI (*g* = −0.17; *p* = 0.002) and nominally lower BW compared to controls (*g* = −0.06; *p* = 0.439) at the post-intervention time-point. In veterans, BW was nominally lower in those given the intervention, although this failed to reach significance (*g* = −0.10; *p* = 0.294). Forest plots for each meta-analysis are provided in Appendix A.

### 3.5. Meta-Regression Analyses

Meta-regression analyses were performed to test whether there was a significant relationship between various continuous variables and the effect sizes of BW and BMI in the group that received the weight loss intervention. Results are presented in Table 3. The continuous variables investigated in individual meta-regression analyses were the age of the sample, the duration of the intervention, the BW at baseline, and the BMI at baseline. None of these variables were significantly associated with the difference in BW or BMI between pre-intervention and post-intervention values (Table 3). An overview of the studies included in the meta-regression analyses is also provided in Appendix A.

### 3.6. Sensitivity Analyses

All main meta-analyses showed low (BW longitudinal, *I*^2^ = 3.45%; BW cross-sectional, *I*^2^ = 32.7%; BMI cross-sectional, *I*^2^ = 0.0%) or moderate (BMI longitudinal, *I*^2^ = 61.2%) heterogeneity. The Egger’s test for small study effects found that one meta-analysis had likely publication bias (BW cross-sectional: t = −0.18, *p* = 0.859; BMI cross-sectional: t = 0.04, *p* = 0.968; BW longitudinal: t = 1.41, *p* = 0.160; BMI longitudinal: t = 3.41, *p* < 0.001). When adjusting for publication bias using the trim and fill method, the BMI longitudinal meta-analysis remained statistically significant (*g* = −0.20; 95% CI −0.29, −0.11). See Appendix A for funnel plots.

Regarding the meta-analyses per sample type, low to moderate heterogeneity was also observed for both active-duty military personnel (BW longitudinal, *I*^2^ = 12.6%; BW cross-sectional, *I*^2^ = 31.1%; BMI longitudinal, *I*^2^ = 64.3%; BMI cross-sectional, *I*^2^ = 0.00%) and veterans (BW longitudinal, *I*^2^ = 4.71%; BW cross-sectional, *I*^2^ = 47.6%). The Egger’s test for small study effects revealed that two meta-analyses had potential publication bias for active-duty military personnel (BW longitudinal: t = −2.27 *p* = 0.023; BMI longitudinal: t = −2.90, *p* = 0.004; BW cross-sectional: t = 0.24, *p* = 0.813; BMI cross-sectional: t = 0.10, *p* = 0.917) and veterans (BW longitudinal: t = 1.13 *p* = 0.257; BW cross-sectional: t = −0.94, *p* = 0.345). The Duval and Tweedie trim and fill method was used to identify funnel plot asymmetry and adjust to publication bias for both active-duty military (BW longitudinal: *g* = −0.07; CI −0.17, 0.02; BMI longitudinal: *g* = −0.16; CI −0.35, 0.03; BW cross-sectional: *g* = −0.07; CI −0.18, 0.04; BMI cross-sectional: *g* = −0.18; CI −0.29, −0.08) and veterans meta-analyses (BW longitudinal: *g* = −0.17; CI −0.32, −0.01; BW cross-sectional: *g* = −0.05; CI −0.16, 0.06). Funnel plots for each meta-analysis per sample type are provided in the Appendix A.

### 3.7. Results for Separate Meta-Analyses Divided by Intervention Type

Alongside the evaluation of the primary outcome of the study, additional meta-analyses were conducted with the aim of evaluating the effectiveness of the different types of interventions included in the study. The main intervention sub-groups were summarized as follows: behavioral and lifestyle intervention, diet and nutritional intervention, self-monitoring intervention, counseling-provided intervention, and Internet-based intervention. Pre-to-post BW and BMI values for the intervention group were analyzed both longitudinally and cross-sectionally compared to control at the post-intervention timepoint for each kind of intervention. The main findings showed that all the intervention subtypes showed a statistically significant decrease in BMI values for both longitudinal and cross-sectional meta-analyses, except for the counseling cross-sectional meta-analysis (Table 4). A comprehensive overview of all the results (number of studies included, Hedge’s g, Confidence Intervals, *p*-values) for each meta-analysis is provided in Appendix A.

### 3.8. Narrative Synthesis of Additional findings

The results of studies excluded from the meta-analysis (n = 5) were narratively synthesized (see Appendix A). The main findings were that behavioral interventions consisting of classes and individual nutritional counseling with a dietitian, as well as lifestyle change promotion using a pedometer, respectively, resulted in a greater and statistically significant decrease in average waist circumference, percent of body fat, and BMI in the treatment group compared to controls [73], and a significant change from baseline to post-intervention in BW, BMI, and percent of body fat [86].

However, an examination of the effectiveness of a second-year weight loss intervention based on behavioral modifications with non-clinician life-style coach sessions compared to usual care showed a significant weight regain in the participants, highlighting how the weight loss intervention was not sufficient to sustain the initial weight loss achieved during the follow-up [79].

## 4. Discussion

The present study is the first meta-analysis and systematic review of RCTs investigating the effectiveness of weight loss interventions to treat overweight and obesity in military populations, comparing the intervention group both longitudinally and cross-sectionally with controls. Our findings highlight how military personnel, such as active-duty military soldiers as well as veterans enrolled in weight loss intervention programs, achieve an overall body weight and BMI reduction from baseline to post-intervention. Moreover, at the post-intervention time-point, BMI was lower than controls with a small-to-moderate effect size. The effectiveness of the weight loss interventions was also found to be significant largely in active-duty military personnel but not in veterans. These findings indicate that weight management interventions in the military population are effective, updating previous research [20,65,89]. Most of the included interventions were based on behavior change strategies adopting a comprehensive approach with different therapeutic goals, combining lifestyle modification with diet and nutritional changes and physical activity therapy, in line with previous evidence [20]. 

The limited role of pharmacotherapy in this population reflects current practice [90]. Indeed, only one RCT study comparing Orlistat versus Placebo has been identified through screening and was included in the meta-analyses. However, it may be that the introduction of new medications such as Glucagon-Like Peptide-1 (GLP-1) receptor agonists may change practice: the American Association of Clinical Endocrinologist guidelines emphasize this point with consideration of those patients who have not responded to intensive lifestyle therapy, or have experienced weight regain after responding to lifestyle therapy, and those with more severe complications of obesity [91]. Adding a pharmacotherapeutic adjunct to behavioral and lifestyle therapy might turn out to be an effective strategy to stem the rising rate of obesity in the military population, especially for veterans [92], although further studies are needed in this regard.

Another relevant consideration of our research concerns the possible application of new therapeutic approaches based on the use of technology. Several forms of remote, internet-based interventions have been identified by our study, enhancing the increasing evidence that an internet-based weight management program could have relevant clinical implications, especially in the military population [55,93,94,95]. Indeed, among soldiers enrolled in the Army weight control program (AWCP), 29% (28% of males and 36% of females) reported that they would like access to an internet-based program to facilitate their weight loss efforts [96]. From this perspective, internet-based interventions may present a platform for a widespread approach to weight management [97] and offer a possible treatment option for the Navy or Marine Corps while aboard [98], in other cases when face-to-face interventions are not possible (such as during the restrictions due to the COVID-19 pandemic [23]), or to facilitate follow-up over time.

Another relevant consideration of these findings concerns the duration of the intervention: only two studies examined the effectiveness of the intervention after 12 months had passed [79,88]. In recent years, several meta-analyses of weight loss interventions conducted on the general population based on pharmacological (BMI_ES_: −2.37, [99]), cognitive behavioral therapy (BMI_ES_: −0.63; [100]), and lifestyle and dietary interventions (BMI_ES_: −1.5; [101]) have been conducted, which also report successful weight loss induction but poor maintenance of weight loss [102,103]. Strategies used to maintain the weight loss include regular contact with a lifestyle counselor [95] or innovative strategies targeting participants at key moments of disengagement risk [104]. More research is needed to assess the long-term effectiveness of weight management interventions in military populations.

Even though we found that behavioral and lifestyle interventions, diet and nutritional interventions, self-monitoring interventions, counseling-provided interventions, and internet-based interventions are all effective in military populations, these approaches might not work for individual military personnel. For example, long-term face-to-face interventions are not feasible for military personnel with frequent different deployments or for sailors. Research should therefore address the specific needs of people with obesity serving in specific military branches and units. 

Additionally, most studies have used body weight or BMI as outcome parameters to measure the success of weight loss programs. However, soldiers often endeavor to gain muscle mass in order to cope with the physical demands of their job or to appear physically fitter and more muscular, which can even become a mental health problem [105]. In fact, the comparatively smaller effect size of weight loss interventions in this population compared to general populations may be because of the different distribution of muscle mass and fat mass when considering the whole BMI composition in these populations. In this case, body weight and BMI metrics might be misleading indicators of health. Thus, BMI and body weight should not be the only outcomes of weight loss interventions; other parameters such as body fat percentage, waist-to-hip ratio, or abdominal circumference should also be considered [106].

## 5. Clinical Implications and Practical Recommendations

Recommendations for clinical practice can be outlined from the findings of this study (Table 5). First, it is recommended to use as comprehensive an approach as possible, as combining multiple therapeutic targets has been found to be the strategy with the greatest effectiveness. This outcome is in line with the latest update of the U.S. Veterans Health Administration (VHA) and Department of Defense (DoD) Clinical Practice Guidelines for the management of adult overweight and obesity, confirming how comprehensive lifestyle interventions (CLIs) have been, and continue to be, the foundation of weight loss management. CLIs combine three critical elements: nutritional, behavioral, and physical activity modification, with the goal of promoting a negative energy balance [107]. Successful interventions can be delivered with both individual and in-group sessions, but the strongest evidence was found to be for tailored approaches [108]. Indeed, both dietary strategies and physical activities are essential components of achieving weight loss. However, this goal can be achieved through various types of meal replacement and food regimens or different types of training sessions (such as aerobic or resistance/physical activities), and there is no one standardized program that universally represents the most effective approach. Thus, it is important to personalize the treatment strategy according to the patient and his or her medical comorbidities, if any, and to establish a collaborative plan that can also be useful in improving therapeutic adherence. Therefore, a successful approach might be to include the patient in an interdisciplinary approach, comprising a dietitian or nutritionist to deal with diet and food intake, a fitness coach for exercise, and a psychologist to provide psychoeducation to the patient by working on the emotional, cognitive, and social factors that influence their relationship with food and the environmental factors that may have a negative influence on lifestyle. This may have a positive influence on long-term weight management and may increase compliance with the dietary and exercise regimen [109]. 

It must also be considered that the specific military environment can play an important role in dietary and exercise restrictions. While both the Army and the Navy are focused on maintaining physical fitness and weight standards, the unique environments, operational demands, and challenges they face result in tailored approaches to weight loss interventions. The Navy might need to consider dietary and exercise adjustments that account for the specific challenges of being on a ship or naval base for a long period of time. This could involve more focus on balanced nutrition that remains accessible during long journeys and considering the limited space for storage. On the other hand, access to fresh versus ultra-processed foods (UPF) can significantly impact weight loss interventions in those populations. Access to fresh, whole foods is often preferred for weight loss and overall health due to their higher nutritional value and benefits. However, in certain military settings, such as deployments or naval environments, access to fresh food might be limited or impractical. Military interventions focusing on weight loss might need to strike a balance between providing nutrient-dense fresh foods whenever possible and incorporating healthier options within the constraints of the military environment. The next stage might be to develop qualitative studies to address the implications of access to fresh versus UPF in weight loss interventions, emphasizing the importance of adapting dietary guidelines to suit the practical challenges and unique circumstances faced by military personnel in different settings. 

Interesting and promising evidence on the use of technology to deliver lifestyle modifications during a weight loss intervention can also be outlined from the present study. Although this represents a still-emerging topic, the use of web-based intervention and computer remote sessions is certainly a rapidly growing area. Although in-person dietary, physical activity, and behavioral interventions have the strongest evidence of effectiveness, the results of our study showed that in clinical practice, the use of internet-based interventions could be an effective strategy to enable the implementation of weight loss interventions when in-person sessions are not possible (such as due to geographical constraints or military service abroad).

Concerning the role of pharmacotherapy, our research highlighted how the use of medications is poorly employed for short-term weight loss interventions in active military populations. Only one RCT investigating the use of orlistat in addition to lifestyle modification in these populations was found. This reflects current clinical practice, as the use of pharmacological therapy is not the first-line approach, although it can certainly be an important tool for weight loss management in combination with CLIs, especially in the long term. Indeed, a systematic review and meta-analysis including 28 RCTs (n = 29,018) investigating the role of pharmacotherapy for the treatment of obesity in the general population concluded that the prescription of medications (such as liraglutide, naltrexone/bupropion, orlistat, or phentermine/topiramate) is recommended in addition to CLIs, especially for patients with a BMI of ≥30 kg/m^2^, and as a long-term therapeutic strategy [110]. There is little support in the literature for the short-term use of medications for weight loss, and current international guidelines recommend the introduction of medications when CLIs have produced insufficient results [107]. The same applies to the use of bariatric surgery: current U.S. VA/DoD Clinical Practical Guidelines point to bariatric surgery in general populations as a possible long-term treatment strategy in conjunction with CLIs for patients who are less than 65 years old and who have a BMI ≥ 40 kg/m^2^, or patients who have an obesity-associated medical condition(s) with a BMI ≥ 35 kg/m^2^ [107]. There is still a dearth of evidence for the long-term management of weight loss and maintenance interventions in military populations. Current guidelines recommend that if a patient has lost weight in a short-term treatment program, they should be placed in a program for weight maintenance over time [111]. Moreover, adjunctive pharmacotherapy could be combined with CLIs, as has been observed in the general population. Overall, further studies are currently needed to confirm the long-term effects of weight loss interventions in military populations before definitive recommendations can be made.

## 6. Strengths and Limitations

To the best of our knowledge, this systematic review and meta-analysis is the first of its kind to comprehensively review the literature on RCTs to address the effectiveness of weight loss interventions in the military population. Selection bias was mitigated by involving two independent reviewers during the screening, data extraction, and quality assessment procedures. However, several limitations should be highlighted. First, there is wide heterogeneity in the type of intervention, and interventions may have included several active elements. In addition, a separate meta-analysis could not be conducted to investigate the effectiveness of the weight loss interventions in terms of BMI values in the sample of only veterans due to a lack of sufficient data, as well as for gender-specific sub-group analyses. Finally, although inclusion criteria allowed the selection of ongoing studies to maximize inclusion, as well as the selection of articles in different languages (English, German, Italian, Spanish, and French) to limit possible bias due to missing data and ensure a high level of generalizability, most of the articles after the screening process were from the United States.

## 7. Conclusions

This systematic review and meta-analysis found a small to moderate short-term effectiveness of weight loss interventions among the military population and a promising effectiveness of Internet-based programs. However, there is still a need for further research to evaluate the long-term effects and weight loss maintenance following the intervention.

## Figures and Tables

**Figure 1 nutrients-15-04778-f001:**
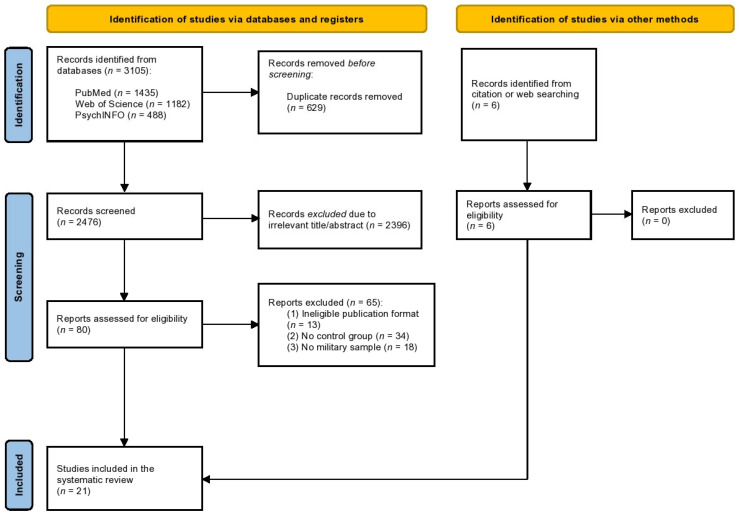
PRISMA flow diagram of the literature screening. Adapted from Page et al., 2021 [66].

**Figure 2 nutrients-15-04778-f002:**
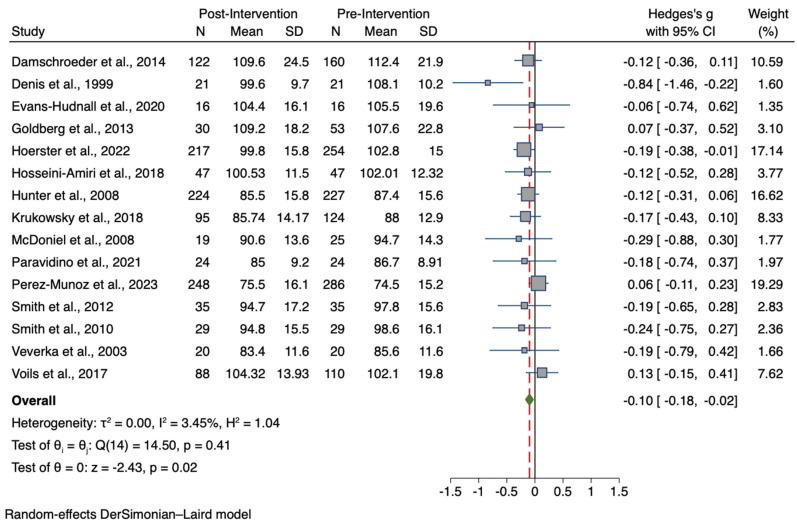
Forest plot of Hedge’s *g* comparing pre-intervention and post-intervention BW values of the treatment group [57,71,72,74,75,76,77,78,80,82,83,84,85,87,88]. Zero indicates no effect, whereas values on the left of this line indicate a decrease in BW when comparing values at baseline and after treatment. The dashed line represents the overall effect size.

**Figure 3 nutrients-15-04778-f003:**
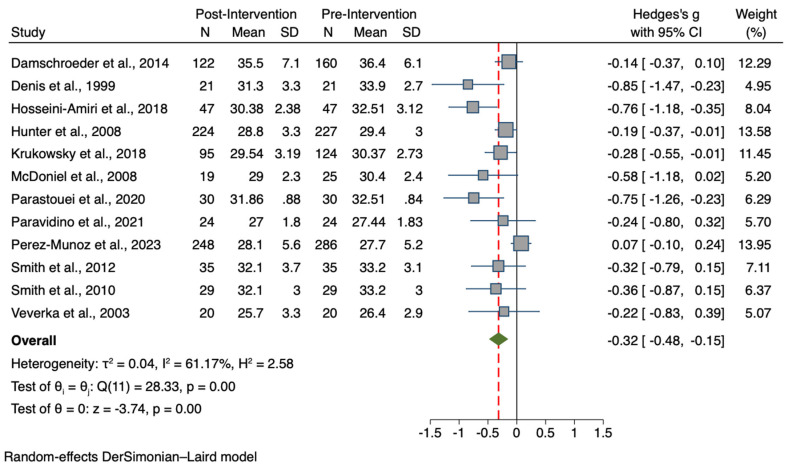
Forest plot of Hedge’s *g* comparing pre-intervention and post-intervention BMI values of the treatment group [57,71,72,77,78,80,81,82,83,84,85,87]. Zero indicates no effect, whereas values on the left of this line indicate a decrease in BMI when comparing values at baseline and after treatment. The dashed line represents the overall effect size.

**Figure 4 nutrients-15-04778-f004:**
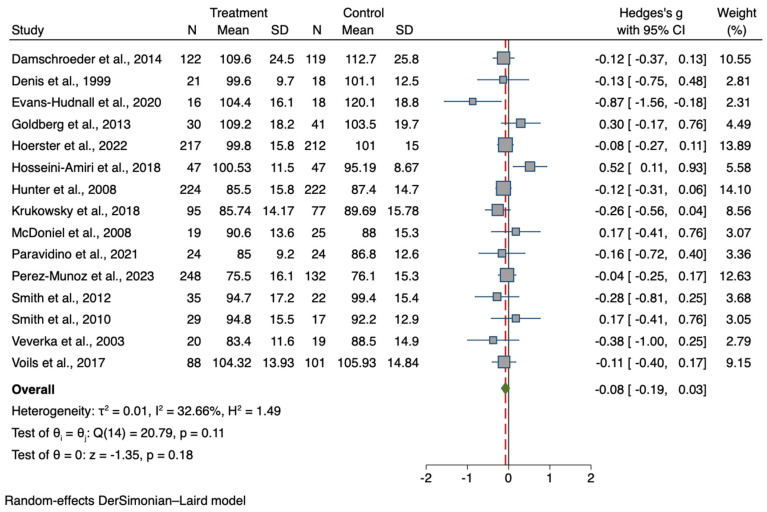
Forest plot of Hedge’s *g* of BW values in the treatment group compared to controls [57,71,72,74,75,76,77,78,80,82,83,84,85,87,88]. Zero indicates no effect, whereas values on the left of this line indicate a decrease in body weight when comparing values between the treatment and control groups at the post-intervention time-point. The dashed line represents the overall effect size.

**Figure 5 nutrients-15-04778-f005:**
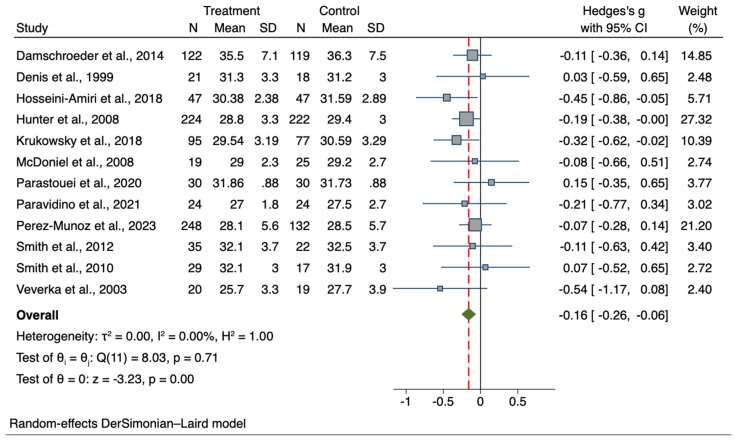
Forest plot of Hedge’s *g* of BMI values in the treatment group compared to controls [57,71,72,77,78,80,81,82,83,84,85,87]. Zero indicates no effect, whereas values on the left of this line indicate a decrease in body mass index when comparing values between the treatment and control groups at the post-intervention time-point. The dashed line represents the overall effect size.

**Table 1 nutrients-15-04778-t001:** Study and sample characteristics for studies included in the systematic review and meta-analysis.

Study, Country	Year	Population	Sample (N)	Age (Years, M ± SD)	Study Design	Intervention	BW (M ± SD) T0 T1	BMI (M ± SD) T0 T1	Duration (Weeks)
Afari et al., [60] USA (Study ongoing)	2019	US Navy	178	29.7 ± 6.9	RCT	ACT + SS	94.8 ± 18.8 NR	33.1 ± 3.9 NR	8
Boutelle et al., [70] USA (Study ongoing)	2023	US Veterans	129	47.1 ± 11.3	RCT	CHARGE	NR NR	34.8 ± 4.7 NR	20
Damschroder et al., [71] USA	2014	US Veterans	481	55.0 ± 10.0	RCT	ASPIRE	113.2 ± 23.2 111.1 ± 25.1	36.6 ± 6.2 35.9 ± 7.3	52
Dennis et al., [72] USA	1999	US Navy	39	31.2 ± 6.5	RCT	Shipboard Weight Control Program	107.5 ± 11.0 100.3 ± 11.0	33.5 ± 2.8 31.2 ± 3.2	26
Erickson et al., [73] USA	2017	US Veterans	121	51.3 ± 9.2	RCT	LB Intervention	103.1 ± NR 101.8 ± NR	NR NR	52
Evans-Hudnall et al., [74] USA	2020	US Veterans	34	58.7 ± 9.1	RCT	HERO	112.8 ± 23.0 112.7 ± 17.5	36.7 ± 7.0 NR	16
Goldberg et al., [75] USA	2013	US Veterans	109	52.0 ± 9.1	RCT	MOVE!	106.0 ± 21.7 105.9 ± 19.1	NR NR	26
Hoerster et al., [76] USA	2022	US Veterans	511	57.4 ± 13.9	RCT	D-ELITE	102.3 ± 14.5 100.4 ± 15.4	NR NR	52
Hosseini-Amiri et al., [77] Iran	2018	Active Soldiers	94	23.3 ± 1.6	RCT	EPPM	100.1 ±10.7 97.9 ±10.1	31.9 ± 2.7 30.9 ± 2.6	4
Hunter et al., [57] USA	2008	US Air Force active-duty personnel	446	34.0 ± 7.3	RCT	BIT	87.0 ± 15.2 86.4 ± 15.3	29.3 ± 3.0 29.1 ± 3.1	26
Krukowski et al., [78] USA	2018	Active duty-military personnel	248	34.6 ± 7.5	RCT	Look AHEAD ILI	89.0 ± 14.3 87.5 ± 14.9	30.4 ± 2.9 29.9 ± 3.2	52
Lutes et al., [79] USA	2017	US Veterans	332	55.9 ± 9.5	RCT	ASPIRE-SC	113.0 ± 22.4 111.4 ± NR	36.2 ± 6.0 NR	104
McDoniel et al., [80] USA	2008	US Air Force active-duty personnel	54	28.0 ± 7.3	RCT	“Sensible Weight” Program	90.5 ± 14.4 89.1 ± 14.6	29.8 ± 2.4 29.1 ± 2.5	13
Parastouei et al., [81] Iran	2020	Military Personnel	60	41.5 ± 7.2	RCT	Synbiotic Supplementation	NR NR	32.1 ± 0.8 31.8 ± 0.9	8
Paravidino et al., [82] Brazil	2021	Military trainer of Naval Academy	72	21.0 ± 2.0	RCT	EFECT study	87.3 ± 9.6 86.4 ± 10.2	27.9 ± 2.1 27.6 ±2.1	2
Perez-Munoz et al., [83] USA	2023	Active-duty Military Women and TRICARE beneficiaries	430	30.6 ± 4.9	RCT	PPWL Intervention	74.2 ± 15.0 74.7 ± 15.0	27.6 ± 5.2 28.2 ± 5.6	26
Smith et al., [84] USA	2010	US Army Soldiers	113	28.4 ± 7.4	RCT	Meal-Replacement Program	97.2 ± 15.1 93.8 ± 15.5	32.8 ±3.0 32.0 ± 3.0	26
Smith et al., [85] USA	2012	Active-duty Soldiers	435	NR	RCT	Orlistat	99.6 ± 15.8 96.5 ± 16.5	33.3 ± 3.4 32.3 ± 3.7	26
Staudter et al. [86] USA	2011	US Active-duty military	106	50.0 ± 9.3	RCT	Pedometer Intervention	87.6 ± 16.3 NR	32.5 ± 5.4 NR	12
Veverka et al., [87] USA	2003	US Air Force active-duty personnel	39	NR	RCT	Stages of Change Model	85.5 ± 12.9 85.9 ± 13.2	26.9 ± 3.3 26.7 ± 3.6	26
Voils et al., [88] USA	2017	US Veterans	222	61.8 ± 8.3	RCT	Maintenance Intervention	103.6 ± 20.4 105.2 ± 14.4	34.0 ± 6.1 NR	56

Abbreviations: BW: body weight; BMI: body mass index; T0: baseline time-point (pre-intervention); T1: post-intervention time-point; RCT: randomized controlled trial; ACT: Acceptance and Commitment Therapy; SS: ShipShape; NR: Not Reported; CHARGE: Controlling Hunger and ReGulating Eating for Veterans; ASPIRE-SC: The Aspiring for Lifelong Health program-Small Changes intervention; LB: Lifestyle Balance; HERO: Healthy Emotions and Improving Health Behavior Outcomes; D-ELITE: Evaluation of Lifestyle Interventions to Treat Elevated Cardiometabolic Risk in Primary Care; EPPM: Extended Parallel Process Model on knowledge, attitudes, and practices; BIT: Behavioral Internet Therapy; ILI: Intensive Lifestyle Intervention; EFECT: Physical Exercise and Compensatory Effects; TRICARE: The uniformed services health care program for active-duty service members and active provided by the United States Department of Defense (DoD); PPWL: Post-partum Weight Loss.

**Table 2 nutrients-15-04778-t002:** Summary of overall outcomes and heterogeneity for cross-sectional and longitudinal BW and BMI meta-analyses, with analyses according to sample type (active-duty personnel and veterans).

*Group*	*N*	*SMD*	*95% CI*	*Z*	*p*	*Heterogeneity*
**Pre-to-post intervention**	(Pre, Post)					
*Overall*						
BW (n = 15)	1431, 1235	−0.10	−0.18, −0.02	−2.43	0.015 *	*I*^2^ = 3.45%; *p* = 0.413
BMI (n = 12)	1028, 914	−0.32	−0.48, −0.15	−3.74	<0.001 *	*I*^2^ = 61.2%; *p* < 0.001 **
*Active-duty personnel*						
BW (n = 10)	838, 726	−0.12	−0.23, −0.00	−2.04	0.041 *	*I*^2^ = 12.6%; *p* = 0.327
BMI (n = 11)	868, 792	−0.35	−0.54, −0.16	−3.62	<0.001 *	*I*^2^ = 64.3%; *p* <0.002 **
*Veterans*						
BW (n = 5)	593, 476	−0.09	−0.21, −0.04	−1.36	0.174	*I*^2^ = 4.71%; *p* = 0.380
**Treatment vs. controls**	(Treatment, Control)					
*Overall*						
BW (n = 15)	1235, 1094	−0.08	−0.19, 0.03	−1.35	0.178	*I*^2^ = 32.7%; *p* = 0.107
BMI (n = 12)	914, 752	−0.16	−0.26, −0.06	−3.23	0.001 *	*I*^2^ = 0.00%; *p* = 0.711
*Active-duty personnel*						
BW (n = 10)	762, 603	−0.06	−0.21, 0.09	−0.77	0.439	*I*^2^ = 31.1%; *p* = 0.159
BMI (n = 11)	792; 633	−0.17	−0.27, −0.06	−3.14	0.001 *	*I*^2^ = 0.00%; *p* = 0.643
*Veterans*						
BW (n = 5)	473, 491	−0.10	−0.30, −0.09	−1.05	0.294	*I*^2^ = 47.6%; *p* = 0.106

Notes. * Significant findings at *p* < 0.05; ** Significant findings at *p* < 0.01; BW: body weight; BMI: body mass index. N: Number of participants; SMD: Standardized Mean Difference; CI: Confidence Intervals; Z: Z-value.

**Table 3 nutrients-15-04778-t003:** Results of meta-regression analyses.

*Group*	*Variable*	*N Studies Included*	*β (SD)*	*95% CIs*	*p*
Intervention Group					
** *BW* **					
	Age	13	−0.002 (0.004)	−0.009, 0.005	0.609
	BW at baseline	15	0.005 (0.003)	−0.001, 0.011	0.135
	Duration of the intervention	15	−0.000 (0.003)	−0.005, 0.005	0.944
** *BMI* **					
	Age	10	0.002 (0.021)	−0.028, 0.015	0.545
	BMI at baseline	12	0.034 (0.028)	−0.021, 0.089	0.224
	Duration of the intervention	12	−0.008 (0.005)	−0.018, 0.003	0.142

Notes. BW: body weight; BMI: body mass index; N: Number; β (SD): Beta Coefficient (Standard Deviation); CIs: Confidence Intervals; *p*: *p*-value (significant findings at *p* < 0.05).

**Table 4 nutrients-15-04778-t004:** Results for longitudinal and cross-sectional BMI meta-analyses divided by intervention.

Type of Intervention	BMI Longitudinal Meta-Analyses				BMI Cross-Sectional Meta-Analyses			
	*N*	*Hedge’s g*	*95% CI*	*p*	*N*	*Hedge’s g*	*95% CI*	*p*
Behavioral and Lifestyle	10	−0.28	−0.45, −0.11	0.001 **	10	−0.18	−0.28, −0.08	< 0.001 **
Diet and Nutritional	9	−0.30	−0.50, −0.11	0.002 **	9	−0.15	−0.26, −0.03	0.010 *
Self-Monitoring	7	−0.26	−0.48, −0.04	0.021 *	7	−0.17	−0.28, −0.06	0.003 **
Counseling Provided	6	−0.30	−0.47, −0.14	0.001 **	6	−0.13	−0.29, 0.03	0.113
Internet-Based	3	−0.22	−0.37, −0.07	0.004 **	3	−0.25	−0.40, −0.09	0.002 **

Notes. * Significant findings at *p* < 0.05; ** Significant findings at *p* < 0.01; N = number of studies; g = Hedge’s g Effect Size; 95% CI: Confidence Interval; *p* = *p*-value.

**Table 5 nutrients-15-04778-t005:** A summary of the main clinical implications and practical recommendations for military populations based on the literature review.

Topic	Clinical Recommendations	Practical Implications	Level of Evidence	RCTs (n)
Short-term weight loss intervention for obesity (up to 6–12 months).	Individual or group-based comprehensive lifestyle intervention	Physical activity (aerobics, resistance, or high intensity); no sufficient evidence from RCTs regarding the superior effectiveness of one type, frequency, or intensity of physical activity.	High	18
		Dietary and nutritional interventions such as meal replacements promoting low caloric balance intake and healthy meal plans provided by a registered dietitian (when available) and individualized to each patient.	High	12
		Cognitive behavioral therapy, psychoeducational strategies, and motivational techniques for cognitive, emotional, and social factors that influence weight management.	High	12
		Structured outcome monitoring over time (clinical or self-monitoring): body weight, BMI, fat percentage, waist-to-hip ratio, abdominal circumference.	High	12
		Internet-based intervention when in-person programs are not available.	Good	5
		Behavioral therapy plus the use of technology (e.g., pedometer).	Weak	2
		Pharmacological intervention (e.g., Orlistat).	Weak	1
Long-term weight loss intervention for obesity	Military personnel who have lost weight should be enrolled in a comprehensive weight loss maintenance program.	Lack of evidence for weight maintenance programs in military populations.	Weak	2

Abbreviations: RCTs: randomized controlled trials; n: number; BMI: Body mass index. Algorithm: The level of evidence was evaluated based on the number of available RCTs, rated as follows: “weak” for RCTs n ≤ 2; “good” for RCTs n = 3–5; “high” for RCTs n > 5.

## Data Availability

Not applicable.

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
