# Peer review of "Randomized Controlled Trials to Treat Obesity in Military Populations: A Systematic Review and Meta-Analysis"

_nutrients, 2023, doi:10.3390/nu15224778_

Round 1

Reviewer 1 Report

Comments and Suggestions for Authors

The presented research is very relevant and practically significant, since in the current situation of an increase in obesity among the population three times over the last half century, it is necessary to introduce effective technologies to solve this problem. Today obesity is a systemic disease impacting the whole well-being of the person with consequences on both physical and mental health. Based on a large survey study of experimental work on the problem of obesity in military personnel, the authors assessed the effectiveness of the therapeutic intervention in terms of weight loss, comparing the pre-intervention body weight and BMI values of participants who received the treatment to the postintervention values, as well as comparing the intervention group to controls.

The study has important scientific novelty, as it represents the first meta-analysis and systematic review of RCTs investigating the effectiveness of weight loss interventions to treat overweight and obesity in military populations, comparing intervention group both longitudinally, and cross-sectionally with controls. Based on the comparative experiment of the studies studied, the authors draw an important conclusion: behavioral and lifestyle interventions, diet and nutritional interventions, self-monitoring interventions, counselingprovided interventions, and internet-based interventions are all effective in military populations, however, what is most important these approaches might not work for the individual military personnel. In this case, body weight and BMI metrics might be a misleading indicator of health. In this regard, the authors make one of the main practical recommendations: BMI and body weight should not be the only outcome for weight loss interventions, but other parameters such as body fat percentage, waist-to-hip ratio or abdominal circumference should also be considered.

However, despite the extensive theoretical review work, the presented research would be more relevant and meaningful for practice if the practical recommendations were deeper and more systematic.

Author Response

Dear reviewer,

Thank you for providing your feedback and comments regarding our study. We found them very accurate and helpful in order to make the article more meaningful.

Regarding your suggestion to provide more systematic recommendations for practice, we decided to add a paragraph to the manuscript (see “Section 5. Clinical implications and practical recommendations”) to expand and specify the clinical implications and provide a graphical summary of the main practical recommendations that emerged from our study, as well as integrating them with current international guidelines on the topic.

Thank you very much for your review,

Best wishes,

Davide Gravina

Reviewer 2 Report

Comments and Suggestions for Authors

Thank you for this topic of weight reduction RCTs in a special group of persons. Military persons are in a special situation such as uniformed lifestyle and so on.

Have the authors looked for sex specific effects?

Would it be possible to divided the studies sex specific?

What was the fact of self-reported and measured body weight and BMI?

Regarding is a different time frames of the intervention studies have for our sauce approved which effect loss depending on this timeframe? 

What are the conclusion of the authors for specific interventions in military personnel for body weight of BMI loss?

Comments on the Quality of English Language

Minor corrections in the quality of English are necessary.

Author Response

Dear reviewer,

Thank you for providing your feedback and comments regarding our study, we found them very accurate and helpful in order to make the article more meaningful.

Regarding sex-specific differences in the outcome of the weight loss intervention, we fully agree that it could represent a relevant topic to be investigated. However, we screened all the articles included in the meta-analysis and unfortunately there is not sufficient data provided per gender to perform a subgroup analysis based on gender. However, we decided to add this as a limitation of the study (see “Section 6. Strengths and limitations”).

Regarding the self-reported and measured BW and BMI we added the Table S3 specifying the strategy adopted for each study, and we discussed it (See last paragraph of the Discussion) stating how BMI and body weight should not be the only outcome for weight loss interventions, but other parameters such as body fat percentage, waist-to-hip ratio or abdominal circumference should also be considered as emerged from the literature search.

Regarding your two additional comments concerning timeframe and conclusion about specific interventions, we decided to add a paragraph to the manuscript (see “Section 5 of the latest version of the manuscript”) to further detail the clinical implications and provide a summary of the main practical recommendations.

Regarding the minor corrections in the quality of English, the manuscript has been checked again and edited by co-authors of the paper who are native English language speakers.

Thank you very much for your review,

Best wishes,

Davide Gravina